# Subspecific Nomenclature of *Giardia duodenalis* in the Light of a Compared Population Genomics of Pathogens

**DOI:** 10.3390/pathogens12020249

**Published:** 2023-02-03

**Authors:** Michel Tibayrenc

**Affiliations:** Maladies Infectieuses et Vecteurs Écologie, Génétique, Évolution et Contrôle, MIVEGEC (IRD 224-CNRS 5290-UM1-UM2), Institut de Recherche pour le Développement, CEDEX 5, 34394 Montpellier, France; michel.tibayrenc@ird.fr

**Keywords:** clonality, genetic recombination, parasite, speciation, taxonomy

## Abstract

Genetic and genomic data have long recognized that the species *Giardia duodenalis* is subdivided into at least eight genetic clusters that have been named “assemblages” by specialists in the field. Some of these assemblages have been given the status of species, with Linnean binames. In the framework of the predominant clonal evolution model (PCE), we have shown that, from an evolutionary point of view, *G. duodenalis* assemblages are equatable to “near-clades”, that is to say: clades whose discreteness is somewhat clouded by occasional genetic exchange, but remain discrete and stable in space and time. The implications of this evolutionary status for the species described within *G. duodenalis* are discussed in light of the most recent genetic and genomic studies. The pattern of this species’ subspecific genetic variability and genetic clustering appears to be very similar to the ones of various parasitic, fungal and bacteria species. This underlines the relevance of a compared population genomics of pathogenic species allowed by the broad framework of the PCE model.

## 1. Introduction

*Giardia duodenalis* is a cosmopolitan parasite that causes waterborne infections of the digestive tract called Giardiasis. It infects humans and many other mammalian species, including dogs, cats, rats, beavers and cattle [1]. The medical and veterinary relevance of this parasite is considerable since it can be the cause of serious health problems. Moreover, it is highly transmissible, which explains why molecular epidemiology* (* = terms and abbreviations defined in the Appendix A) approaches have been developed for it for a very long time.

## 2. A Brief Reminder on *G. duodenalis* Genetic Variability

Genetic analysis of *G. duodenalis* is complicated by the fact that this parasite features two nuclei. Their genomes are similar but not identical. This leads to a phenomenon termed “allelic sequence heterozygosity” (ASH).

Since the 80s, *G. duodenalis* has been the target of isoenzyme*/Multilocus Enzyme Electrophoresis* (MLEE*) pioneering studies [2]. These studies have led to suspect that *G. duodenalis* is a complex of species [2]. Further studies were based on Multilocus Sequence Typing* (MLST*) routinely involving three housekeeping genes, namely ß-Giardin (*Bg*), Glutamate Dehydrogenase (*Gdh*) and Triose Phosphate Isomerase (*Tpi*). They all have shown that this species is subdivided into at least eight genetic clusters, labeled A to H, that have been named “assemblages”. These assemblages feature some host specificity [1], although it is not absolute. A and B are zoonoses* and are found in humans and many mammal species. C and D are found in canids; E is isolated from hoofed animals; F is specific to cats; G is to rodents; H is to sea mammals [3,4]. More sophisticated MLST approaches and genomic studies based on whole genome sequencing* (WGS*) and single nucleotide polymorphism* (SNP*) [5] have confirmed the validity of *G. duodenalis* assemblages and have revealed additional genetic clusters within them, termed subassemblages [4].

## 3. *G. duodenalis* Mode of Reproduction and Population Structure

In the framework of the “clonal theory of parasitic protozoa”, we have proposed [6,7] that *G. duodenalis* has a clonal population structure, like many other parasitic species. The evidence was based on the observation of repeated ubiquitous Multilocus Genotypes* (MLGs*) at frequencies that do not fit panmictic* expectations, the presence of statistically significant linkage disequilibrium* and the lack of recombinant genotypes. This clonal pattern has been corroborated by other authors [8], with comparable arguments.

The definition of clonality proposed by us [6,7] limits itself to strongly restrained genetic recombination*. It does not deal with any precise cytological or mating processes and only concerns population structure. It, therefore, includes not only mitotic propagation (“classic” clonality) but also self-fertilization*, strong homogamy*/inbreeding* and several cases of parthenogenesis*. This definition of clonality is shared by many, if not most, scientists working on pathogen population structure (see many references and Table 2 in [9]). Conversely, recombination is equated with sexuality by some authors [10]. However, other authors privilege alternative definitions of clonality. Rougeron et al. [11] restrain clonality to the sole mitotic propagation (“classic” clonality) and distinguish it from selfing/inbreeding. Other authors [5,12] equate clonality with genetic monomorphism.

## 4. *G. duodenalis* and the Predominant Clonal Evolution* (PCE*) Model

The advent of high-resolution genetic and genomic surveys (WGS, SNP) has allowed us to refine the clonal evolution model [13]. We have proposed that predominant clonal evolution concerns all the cases where the impact of genetic exchange is unable to erase the predominant clonality pattern, the most typical manifestation of it being the presence of a persistent phylogenetic* signal at all evolutionary scales. We have forged the term “near-clade*” (NC*; [13]) to designate the phylogenetic clusters observed in many pathogen species. The term “clade*” is not appropriate, since it concerns evolutionary lineages that are strictly separated from each other, whereas it is probable that occasional genetic exchange occurs within many pathogen species, and even, among some species. Now the NCs feature some remarkable properties: despite the possible occurrence of occasional mating, they are discrete and highly stable in space and time, within the whole ecogeographical range of the species considered. We have proposed that the *G. duodenalis* assemblages perfectly fit the definition of NC [14]. One should emphasize that the concept of NC is compatible with some limited genetic exchange, provided that it does not disrupt the prevalent pattern of detectable phylogenetic signal and of Multigene Bifurcating Trees (MGBTs) from macro- to micro-evolutionary levels. Such MGBTs are indeed considered by many authors as the most reliable manifestation of restrained recombination and predominant clonality [15,16,17,18,19]. Species phylogenies should be based on multigene surveys. The analysis of isolated genes may not reflect the true evolutionary story of the species under survey [20]. When *G. duodenalis is* concerned, discrepancies between phylogenies inferred from different genes have been considered as evidence for genetic exchange [10,21]. However, (i) If such incongruences are due to genetic exchange, they say nothing about the frequency of these exchanges; (ii) they may have many possible causes but genetic exchange [14], for example, homoplasies, markers having different molecular clocks* [22], or undergoing different selective pressures, or different evolutionary tendencies. Another possible cause of phylogenetic discrepancies is the presence of mixed genotypes in the same isolate [23].

The existence of gene flow among and within the assemblages still is in debate. Ankarklev et al. [4] postulate that genetic recombination is “widespread” within assemblage A, but also, between A and E. Cooper et al. [21] infer that there is “evidence for nonclonal evolution” within the subassemblage A2, a hypothesis that is based on incongruence between trees of individual loci. Thompson and Ash [24] hypothesize that occasional recombination could be observed when the transmission is more frequent. Tsui et al. [5] consider that recombination is observed in *G.duodenalis,* but it does not affect the interpretation of phylogenetic data.

However, if genetic exchange obtains in *G. duodenalis*, it does not affect either the discreteness or stability of the assemblages, in accordance with the statement by Tsui et al. [5]. These properties of the assemblages (NCs) have been corroborated by many independent studies relying on different tools and samplings [3,4,10,14,25,26,27,28,29,30,31].

## 5. Challenging the PCE Model: Pseudo-Speciation vs. Russian Doll Model*

Ramírez and Llewellyn have claimed [32,33] that the clonal model of pathogens is “artefactual” for the following reasons. These authors have postulated that restrained recombination in pathogens is verified only among the main genetic subdivisions (=NCs) of the species under study, whereas genetic exchange is much more frequent within each of these main clusters. This model is very similar to the model of pseudo-speciation proposed by Maynard Smith et al. [34] (see Figure 1).

We have designed [35] the Russian doll model, that challenges Ramírez and Lewellyn’s proposals [32,33]. It postulates that restrained recombination is not limited to the main genetic subdivisions (NCs) of the species considered. It is also verified within these main NCs. (Figure 2).

Such Russian doll patterns (RDPs) have been observed in many species of pathogenic bacteria, yeasts and fungi, and parasitic protozoa (for a recent survey, see [36]). It, therefore, appears that this evolutionary pattern is widespread in pathogenic microorganisms and that the proposal in [32,33] is not corroborated in the species surveyed by us.

When *G. duodenalis is* concerned, subclustering patterns (=lesser NCs), that is to say: RDPs have been repeatedly recorded within the assemblages (=main NCs). Subassemblages or subtypes (=lesser NCs) within assemblage A have been corroborated by several studies and have been termed A I, II and III [4,29,37]. RDPs within assemblage B are less clear [27,29]. However, substructuring within B has been observed too. Lesser NCs within assemblage B have been termed B III and IV [4,37]. Figure 1 in Cacciò et al. [37] clearly shows RDPs within assemblages A and B. Perez Faria et al. [3] have evidenced additional RDPs within the subassemblages AI and II (their Figure 1). Sprong et al. [38] also identified RDPs within AI, II and III, as well as within BIII and IV (Figure 1 and Figure 2). However, bootstrap support within BIII and IV was weak. Tsui et al. [5], with genomic data (WGS and SNPs), fully confirmed the discreteness of assemblages A and B and the subclusteruring (RDPs) within them. However, the intra-assemblage pattern was different between A and B. AI (termed A1 in their article) was considered “highly clonal” (close to monomorphic according to their definition), while the subclustering within B was clearer and strongly linked (although not 100%) to geographical location (Figure 2 and Figure 3). Lastly, Woschke et al. [39], with the MLST protocol proposed in [4], fully confirmed RDPs within both assemblages A and B.

RDPs within *G. duodenalis* assemblages have been therefore fully corroborated by many authors relying on various techniques and different sampling strategies. This ubiquity of MGBTs down to low evolutionary levels does not rule out the possibility of limited genetic exchange within the assemblages. However, it does indicate that PCE is verified within them, which makes it possible to reject for this species the pseudo-speciation model proposed in [32,33].

## 6. Are *G. duodenalis* Assemblages Equatable to Species?

This is a long-open debate among the specialists in the field. The need to design a robust terminology that permits scientific exchanges is compelling [24]. However, Latin binames are not a panacea. The description of new species should be based on solid criteria. It should be recalled that the description of new species in pathogenic microorganisms is “largely used as a label of convenience” and that there is no universal bacterial species concept [12].

Several authors have proposed to consider assemblages A and B as different species, respectively termed *G. duodenalis* and *G. enterica* [24,26]. It has been also proposed to term assemblages C/D, *G. canis*, assemblage E, *G. bovis*, assemblage F, *G. cati*, and assemblage G, *G. simondi* [24].

However, the specialists in the field seem to be undecided about the use of the biological species concept* (BSC*), which states that different species are defined by the lack of genetic exchange among them [40]. More generally, the criteria on which assemblages should be described as species are unclear. Andersson [41] states that genomics* and experimental recombination in *Giardia* and other parasites will make it possible to explore possible signatures of sexual recombination and to describe distinct biological species. In other words, if sex occurs among assemblages, they would not deserve the status of distinct species, which deals with the BSC [40]. Other authors [1,21,23,26] have similar views. Birky’s opinion [25] is that one lacks clear criteria for describing new species within *G.duodenalis*. Capewell et al. [27] propose that a higher genetic resolution would make it possible to know whether the assemblages are “true species”, without making it clear which criteria are needed for describing “true” species.

It is proposed here not to base the description of assemblages as species on the BSC, which is clearly inappropriate for pathogenic microorganisms that undergo only occasional mating. PCE approach does confirm that *G. duodenalis* assemblages are genetically discrete and highly stable in space and time. These properties are more than enough to apply the phylogenetic species concept* (PSC*; [42]) to them. In the case of *G.duodenalis*, as is the case for other pathogens [36], the PSC should be applied in a flexible way, since the genetic separation among assemblages, according to the NC concept, may not be total.

According to the Russian doll model, the number of lesser NCs in *G.duodenalis*, like for other pathogen species, may be virtually unlimited. Available data for *G. duodenalis* is not sufficient to confirm it, but for other species, RDPs and tiny NCs are observed down to extreme microevolutionary* scales (time scales of historical times, and even, recent year: “measurably evolving pathogens” [43]). This is the case for example of the parasite *Leishmania donovani* and the bacteria *Escherichia coli* and *Staphylococcus aureus* [36]. It would therefore be misleading to design new species on the only criterion of individual genetic clustering, which could lead to countless new species. It is recommendable to design pathogen species based on the PSC only in the case where the new taxa have medical or epidemiological relevance. It is up to the specialists in the concerned field to decide whether it is the case. This is what *G. duodenalis* specialists have done with the assemblages. In their case, the BSC, which is poorly handled in micropathogens, should be abandoned in favor of the PSC, since *G. duodenalis* assemblages exhibit both genetic discreteness and epidemiological relevance.

## 7. PCE, Speciation and Aneuploidy* in *G. duodenalis*

It has been inferred [44] that aneuploidy is widespread in *G.duodenalis*, as is the case for the parasitic species *Leishmania* [45] and *Trypanosoma cruzi* [46]. This fact is highly relevant for population genetics and, therefore, for the problem of speciation in *G. duodenalis.* Indeed, some tests proposed to detect genetic exchange in parasites’ natural populations are based on the hypothesis that these parasites are diploid [11]. Aneuploidy makes therefore the interpretation of these tests debatable [13]. Such tests should hence be used quite cautiously for exploring the problem of genetic exchange in *G. duodenalis*. In relation to this, it can be recalled that the BSC [40], based on genetic isolation among putative species, is ill-adapted for exploring speciation in pathogenic microorganisms.

## 8. Conclusion: Towards a Compared Population Genomics of Pathogenic Microorganisms

In the last 20 years, our knowledge of *G. duodenalis* molecular epidemiology, population genetics and evolution has made significant progress. Extensive comparison with various other pathogen species, including bacteria, parasitic protozoa and fungi [36] shows that *G. duodenalis* population structure is not a special case, and, on the contrary, exhibits striking similarities with that of other pathogens. This underlines the relevance of a compared population genomics of pathogenic species allowed by the broad framework of the PCE model.

The existence of at least eight “assemblages” that can be equated to NCs is firmly established, as well as their stability in space and time. The question of genetic exchange among and within the assemblages needs further exploration, however, it is well ascertained that such possible gene flow does not erase the integrity of the assemblages/NCs in the long run.

It is important to insist on the fact that the new species described within *G. duodenalis*, such as *G. enterica*, *G. bovis*, G. *simondi*, etc., [24] are not equatable to biological species [40] like for example horses and donkeys. From an evolutionary point of view, they are NCs. It is probably the same for other *Giardia* species such as *G. microti* or *G. muris*.

In the near future, genomic studies should be actively developed to better address the problems of this parasite’s drug resistance and pathogenicity and to further explore the *G. duodenalis* population structure at a microevolutionary level, for knowing whether *G. duodenalis* pertains to the category of the so-called “measurably evolving pathogens” [43].

## Figures and Tables

**Figure 1 pathogens-12-00249-f001:**
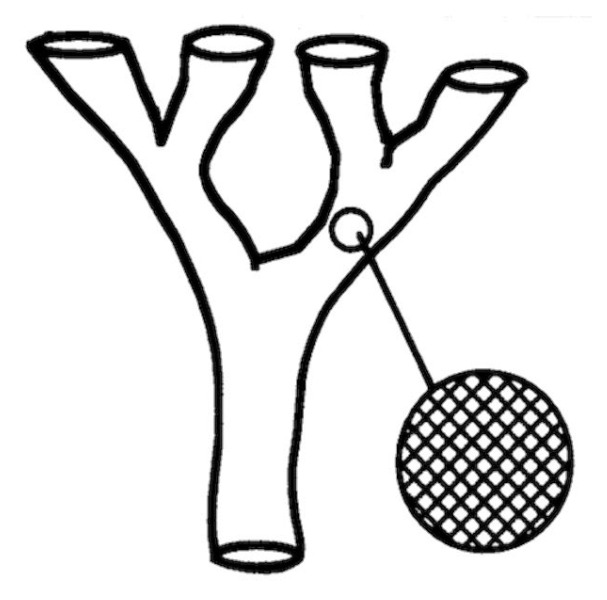
Model of pseudo-speciation: gene flow is inhibited among the main genetic clusters (NCs) of the species under study, whereas it is not within each of these genetic clusters (after Maynard Smith et al. [34]) (Copyright (1993) National Academy of Sciences, U.S.A).

**Figure 2 pathogens-12-00249-f002:**
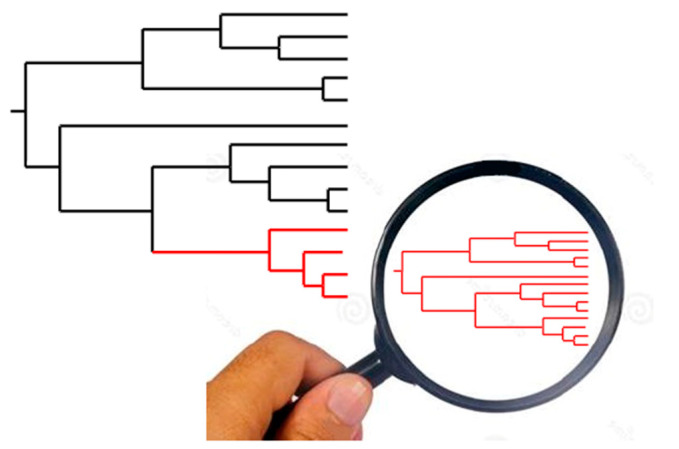
Russian doll model. When population genetic tests are practiced with adequate markers of suitable resolution within each of the NCs that subdivide the species under study (large tree, **left**), they show a miniature picture of the whole species, with the main PCE features, namely, LD and lesser NCs (small tree, **right**). This means that PCE is also verified at these low evolutionary levels.

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
