# Peer review of "Subspecific Nomenclature of Giardia duodenalis in the Light of a Compared Population Genomics of Pathogens"

_pathogens, 2023, doi:10.3390/pathogens12020249_

Round 1

Reviewer 1 Report

In the manuscript entitled “Subspecific nomenclature of Giardia duodenalis in the light of a compared population genomics of pathogens” by Michel Tibayrenc, the author recaps his well-known statement on the “clonal theory” and its consequence for phylogenetic classification in the framework of Giardia duodenalis. The manuscript is very well written and clear. In my view this is a very important contribution/clarification to the overall confusion in the field as to whether name the different “assemblages” different species or not.

The only thing I am missing (in particular for the non-phylogenetic specialists) is an illustration of the PCE and Russian doll model to define the near clades in Giardia phylogenetics. Why are G. duodenalis assemblages near clades and G. muris/G. microti separate species? The representation of the data in the manuscript is very abstract.

Minor:

Line 41: Probably better to use “hoofed animals” instead of “hoofed cattle”

Line 41: specific for cats

line 125: microorganisms

Author Response

Comments and Suggestions for Authors

In the manuscript entitled “Subspecific nomenclature of Giardia duodenalis in the light of a compared population genomics of pathogens” by Michel Tibayrenc, the author recaps his well-known statement on the “clonal theory” and its consequence for phylogenetic classification in the framework of Giardia duodenalis. The manuscript is very well written and clear. In my view this is a very important contribution/clarification to the overall confusion in the field as to whether name the different “assemblages” different species or not.

Answer: thank you for this very positive appreciation.

The only thing I am missing (in particular for the non-phylogenetic specialists) is an illustration of the PCE and Russian doll model to define the near clades in Giardia phylogenetics. Why are G. duodenalis assemblages near clades and G. muris/G. microti separate species? The representation of the data in the manuscript is very abstract.

Answer: Thank you for this very relevant remark. Actually, many “species” described in pathogenic microorganism amount to near-clades in an evolutionary point of view. They received Latin names because the concerned specialists found it relevant. So calling these entities G. muris or G. microti is somewhat arbitrary. Near-clade has a veery sharp evolutionary definition. I developed this important point in the text.

Minor:

Line 41: Probably better to use “hoofed animals” instead of “hoofed cattle”

Answer: OK. Thank you.

Line 41: specific for cats

Answer: OK. Thank you.

line 125: microorganisms

Done

Reviewer 2 Report

This manuscript is an interesting dissertation on the controversy of assemblages or different species in Giardia duodenalis. The author introduces new terms and discusses the usefulness of pseudo-speciation vs. a pattern he named a Russian doll model. Then, it is affordable to hope for a long debate from this manuscript, which would interest the specialized scientific community.

There are only minor recommendations:

The scientific name of the Giardia duodenalis species must be abbreviated as G. duodenalis instead of GD.

Several typos in the manuscript need to be corrected, for example, line 43 (GD}, lines 57-58, etc.

Many asterisks in the manuscript need an explanation of their meaning since the beginning of the manuscript.

It should be better to use the term “allelic sequence heterozygosity” than “allele sequence heterozygosity.” Regarding this, referencing papers as doi: 10.1186/1471-2180-12-65 and 10.1186/s12866-022-02581-3 could help readers.

Since subassemblages are discussed (from line 94), they should be introduced from the section where the eight assemblages are mentioned.

It is up to the author, but “Fractalized patterns” (alluding to the math term fractals) would also be a good term for Russian doll patterns.

Author Response

This manuscript is an interesting dissertation on the controversy of assemblages or different species in Giardia duodenalis. The author introduces new terms and discusses the usefulness of pseudo-speciation vs. a pattern he named a Russian doll model. Then, it is affordable to hope for a long debate from this manuscript, which would interest the specialized scientific community.

There are only minor recommendations:

The scientific name of the Giardia duodenalis species must be abbreviated as G. duodenalis instead of GD.

Answer: OK. Done.

Several typos in the manuscript need to be corrected, for example, line 43 (GD}, lines 57-58, etc.

Answer: corrected

Many asterisks in the manuscript need an explanation of their meaning since the beginning of the manuscript.

Answer: as indicated at the start of the manuscript, terms with an asterisk correspond to the specialized terms and abbreviations listed in the glossary.

It should be better to use the term “allelic sequence heterozygosity” than “allele sequence heterozygosity.” Regarding this, referencing papers as doi: 10.1186/1471-2180-12-65 and 10.1186/s12866-022-02581-3 could help readers.

Answer: done.

Since subassemblages are discussed (from line 94), they should be introduced from the section where the eight assemblages are mentioned.

Answer: done

It is up to the author, but “Fractalized patterns” (alluding to the math term fractals) would also be a good term for Russian doll patterns.

Answer: “Fractalized patterns” would be relevant. However, ““Russian doll patterns” has been introduced and cited in many papers since 2013. Authors grew used to it as they did for “near-clades”. It would be unwise now to upset this with a new term.